# Cellular preservation of musculoskeletal specializations in the Cretaceous bird *Confuciusornis*

Baoyu Jiang[1,2], Tao Zhao[1], Sophie Regnault[3], Nicholas P. Edwards[4], Simon C. Kohn[5], Zhiheng Li[6], Roy A. Wogelius[4], Michael J. Benton[5] & John R. Hutchinson[3]

The hindlimb of theropod dinosaurs changed appreciably in the lineage leading to extant birds, becoming more 'crouched' in association with changes to body shape and gait dynamics. This postural evolution included anatomical changes of the foot and ankle, altering the moment arms and control of the muscles that manipulated the tarsometatarsus and digits, but the timing of these changes is unknown. Here, we report cellular-level preservation of tendon- and cartilage-like tissues from the lower hindlimb of Early Cretaceous *Confuciusornis*. The digital flexor tendons passed through cartilages, cartilaginous cristae and ridges on the plantar side of the distal tibiotarsus and proximal tarsometatarsus, as in extant birds. In particular, fibrocartilaginous and cartilaginous structures on the plantar surface of the ankle joint of *Confuciusornis* may indicate a more crouched hindlimb posture. Recognition of these specialized soft tissues in *Confuciusornis* is enabled by our combination of imaging and chemical analyses applied to an exceptionally preserved fossil.

[1] School of Earth Sciences and Engineering, Nanjing University, Nanjing 210023, China. [2] State Key Laboratory of Palaeobiology and Stratigraphy, Nanjing Institute of Geology and Palaeontology, Chinese Academy of Sciences, Nanjing, 210008, China. [3] Structure and Motion Laboratory, Department of Comparative Biomedical Sciences, The Royal Veterinary College, University of London, Hatfield, Hertfordshire AL9 7TA, UK. [4] School of Earth and Environmental Sciences, University of Manchester, Manchester M13 9PL, UK. [5] School of Earth Sciences, University of Bristol, Bristol BS8 1TH, UK. [6] Key Laboratory of Vertebrate Evolution and Human Origins of Chinese Academy of Sciences, Institute of Vertebrate Paleontology and Paleoanthropology, Chinese Academy of Sciences, Beijing 100044, China. Correspondence and requests for materials should be addressed to B.J. (email: byjiang@nju.edu.cn) or to J.R.H. (email: jhutchinson@rvc.ac.uk).

Birds evolved remarkable changes in their form, function, behaviour and other biological aspects after their origin from theropod dinosaur ancestors in the Jurassic period. Much as flight evolved after its origin in this lineage, the bipedal posture of birds and their ancestors seems to have evolved. Reduction of the tail and its musculature as well as shifts in the body's centre of mass and enlargement of the pectoral limbs are consistent with the inference that birds adopted a more crouched (flexed) limb pose across their evolution; a trend that began within Theropoda[1–3]. Important questions such as how did particular taxa stand and move (and thus what was the timing and tempo of this postural evolution), or what anatomical, biomechanical and physiological mechanisms contributed to these functions, remain daunting. A major problem is the absence of direct evidence of the structure of many soft tissues that would have influenced pelvic limb function[4].

Here, we document a fossil of an Early Cretaceous bird, *Confuciusornis sanctus*, which has some strikingly well-preserved soft tissues around its ankle joint. Microscopic analyses of these tissues indicate that they include tendons or ligaments, fibrocartilages and articular cartilages, with microstructure evident at the cellular level. Further chemical analyses reveal that even some of the original molecular residues of these soft tissues may remain, such as fragments of amino acids from collagen, particularly in the fibrocartilage. This concurs with accruing evidence that some biomolecules may survive, under exceptional circumstances, over many millions of years[5–12]. Our reconstruction of the soft tissues around the ankle joint in this *Confuciusornis* specimen leads to the conclusion that *Confuciusornis* had evolved a more derived form and, presumably, function of the ankle region (for example, neomorphic, rudimentary tibial cartilage and hypotarsus), revealing new details of how early birds began to adopt more crouched hindlimb postures.

## Results

**Specimen discovery and morphology.** Exceptional soft tissues were discovered in an otherwise unprepared *Confuciusornis* specimen (MES-NJU 57002, Museum of Earth Sciences, Nanjing University) that was collected from the most productive *Confuciusornis*-bearing layer of the Yixian Formation in the Sihetun area, Liaoning Province, in NE China, by the first author. The exceptional preservation was enabled by taphonomic processes of charcoalification of carcasses by a hot pyroclastic density flow[13]. Photomicrography, backscatter scanning electron microscopy (BSEM), computed tomography (CT) scanning, X-ray microdiffraction and energy dispersive X-ray spectroscopy (EDX) analyses of the surrounding sediments and long bone mineralizations were described in a prior study[13]. The specimen bears a suite of morphological characteristics that are diagnostic of *Confuciusornis*, such as a straight femoral shaft, ball-shaped femoral head with a distinct capital fossa, straight tibiotarsus, proximally fused and very short tarsometatarsus, slit between metatarsal III and IV proximally, metatarsal I attached distally to the lateral side of metatarsal II, slender and splint-shaped metatarsal V, and highly recurved claws with horny sheaths[14–16] (Supplementary Fig. 1). The proximal and distal tarsals were completely fused to the tibia and metatarsals II–IV, respectively; no sutures are evident in the fully ossified distal tibiotarsus and proximal tarsometatarsus. These skeletal traits indicate that the specimen was skeletally mature[17], even though its body size is perhaps slightly small as indicated by the shorter length of its tarsometatarsus compared with previously published specimens (Supplementary Table 1).

**Histological sectioning and tissue identification.** Two approximately parallel sections oblique to the sagittal plane were prepared from the specimen. Section 1 cuts from the medial surface of the distal tibiotarsus, passing through the medial condyle and a cavity between the tibiotarsus and tarsometatarsus, to the lateral surface of the proximal tarsometatarsus of the right pelvic limb. Section 2 is lateral to section 1, cutting through the medial and lateral condyles, to the lateral surface of the tarsometatarsus (Fig. 1; also see Supplementary Fig. 1b in ref. 13).

Blackened soft tissues with residual, desiccated linear biological structure are exposed intermittently on the plantar surface of the distal tibiotarsus and then across the ankle joint to the proximal tarsometatarsus (Fig. 1f,g). Scanning electron microscopy-based energy-dispersive X-ray analysis shows that the tissues comprise mainly carbonaceous materials[13] (Supplementary Fig. 2). Three types of soft tissues are recognizable.

Tissue type 1 is exposed from the distal tibiotarsus, through the space between the tibiotarsus and tarsometatarsus, to the proximal tarsometatarsus. It is composed of parallel arrays of wavy fibrils forming bundles of 1–10 μm in diameter, separated by linear fissures ('t/l', Figs 2a,b and 3; Supplementary Fig. 3). This tissue bears the hallmarks of tendons or ligaments, such as the wavy appearance, parallel arrangement and hierarchical organization of the fibrils[18,19], distinct from purely sedimentary features or other tissues. Hence, we interpret tissue type 1 as tendon and/or ligament.

Tissue type 2, with thickness up to 0.5 mm, occurs in the areas on the distal tibiotarsus and the proximal tarsometatarsus where the tendons/ligaments would have wrapped around the condyle and cotyle. It consists of oval or round cellular structures 5–10 μm in diameter that are either embedded in a matrix composed of a network of interwoven fibrils and tiny mineralized zones, or arranged in rows between parallel fibres ('fc', Figs 2 and 3a–c). These features are typical of fibrocartilage[20,21], and hence we interpret this tissue's identity as fibrocartilage. The fibrocartilages appear to be locally (at least partly) ossified, as indicated by the presence of small mineral precipitates in the matrix and mineralized cellular structures on the upper surface of the underlying bone ('m', Figs 2c,d and 3a–c).

Tissue type 3 is preserved along both articular facets of the tarsometatarsus and tibiotarsus. It contains two thin layers of densely packed, paired cellular structures in a matrix composed of a network of interwoven fibrils: an inner mineralized layer and an outer unmineralized layer (Fig. 2e,f). The cellular structures are oval or round and about 10 μm in diameter. The structures, resembling the lacunae left by paired cells of articular cartilage in their shape, size and positions of occurrence[22], indicate that the articular cartilages are partly preserved. The mineralized and unmineralized layers correspond respectively to the calcified and uncalcified zones of articular cartilages. This interpretation is supported by gross morphology and histology of the corresponding extant avian tissues (Supplementary Figs 4,5 and 6), and previous reports of cartilage preservation in fossils[23].

**Chemical analyses of possible soft tissues.** Tendons, ligaments and cartilage are mainly composed of collagen and the proteoglycan aggrecan[24]. Survival and detection of residual amide functional groups derived from precursor proteins within fossil specimens is well documented via Fourier transform infrared (FTIR) and time-of-flight–secondary ion mass spectrometry (ToF–SIMS)[5–11,25]. We tested our inferences based on the morphological similarities of these putative soft tissues to tendons, ligaments and fibrocartilages with chemical analyses.

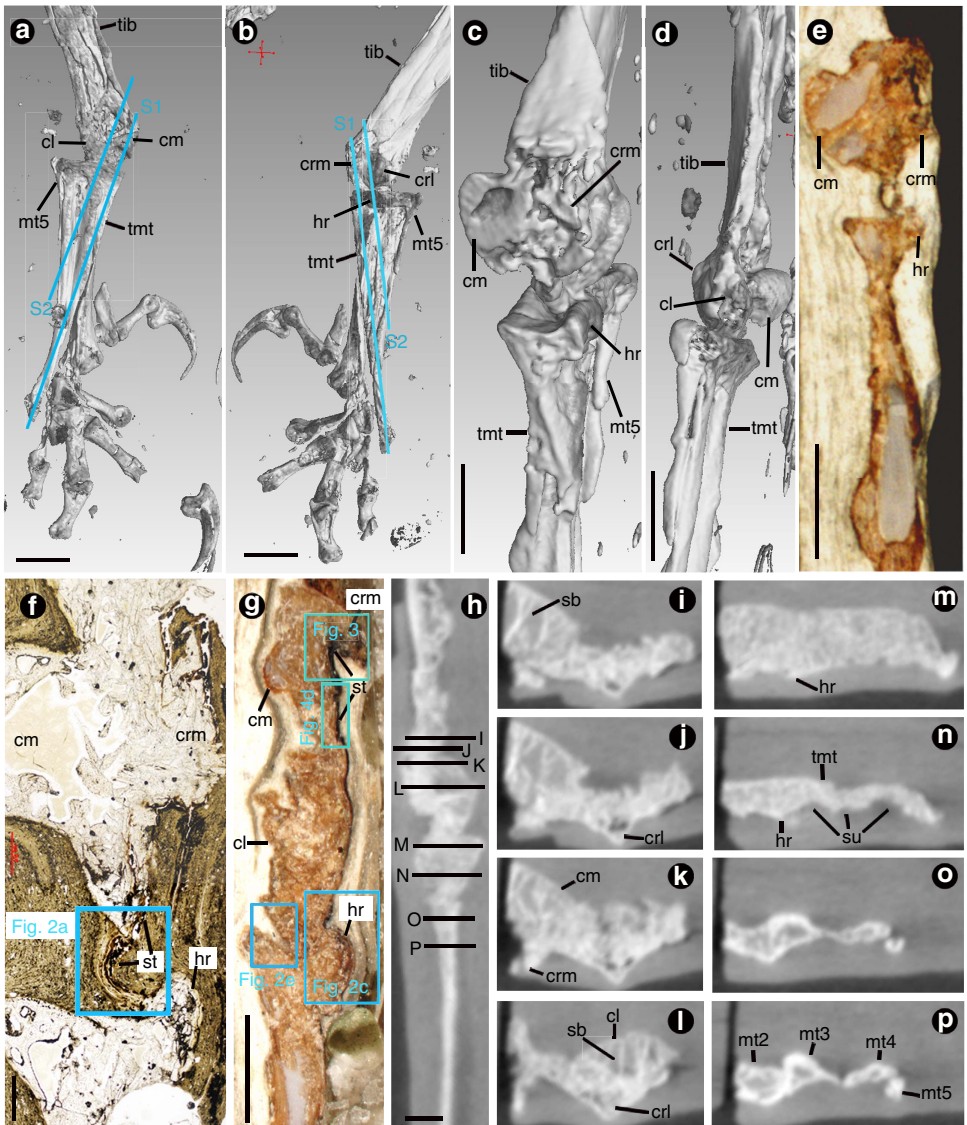

**Figure 1 | Osteology and putative soft tissues of the *Confuciusornis* hindlimb.** Specimen MES-NJU 57002 (right lower limb), showing position of sections. (**a**,**b**) Three-dimensional digital images, constructed from microCT scan data, showing approximately dorsal (**a**) and plantar (**b**) views of the right distal hindlimb and positions of sections. (**c**,**d**) Close-up view of the ankle joint in approximately medial (**c**) and lateral (**d**) views. (**e**–**g**) Photomicrographs of the ankle joint before sectioning (**e**), in section 1 (**f**), and in section 2 (**g**). (**h**) Approximately sagittal microCT slice showing positions of horizontal microCT slices. (**i**–**p**) Continuous horizontal microCT slices across the distal tibiotarsus (**i**–**l**) and proximal tarsometatarsus (**m**–**p**). cl, condylus lateralis; cm, condylus medialis; crl, crista lateralis; crm, crista medialis; hr, hypotarsal ridge; mt, metatarsal; S1/S2, sections 1/2; sb, spongy (cancellous/trabecular) bone; st, soft tissue; su, sulcus; tib, tibiotarsus; tmt, tarsometatarsus. Scale bar, 5 mm in **a**–**e** and **g**, 500 μm in **f**, 1 mm in **h**–**p**.

FTIR analysis was applied to the tissues exposed on the polished section in reflection mode and the spectra were compared with those of modern intact collagen and aggrecan[26]. As shown in Fig. 4a, the fossilized soft tissues have three main strong absorbance regions. The region of 960–1,160 cm$^{-1}$ is strongest with a prominent peak at 1,033 cm$^{-1}$, followed by that of 1,400–1,700 cm$^{-1}$ with two strong peaks at 1,510 and 1,652 cm$^{-1}$, respectively. The region of 1,200–1,300 cm$^{-1}$ is relatively weaker, with a peak at 1,248 cm$^{-1}$. The peaks at 1,652 and 1,510 cm$^{-1}$ correspond to the diagnostic amide I and II absorbances, respectively, which appear distinctly in the spectra of collagen and aggrecan. The presence of the amide I peak at 1,652 cm$^{-1}$ in both fossil spectra, combined with the additional structure in both spectra from 1,652 to 1,400 cm$^{-1}$, indicate that organic material is present within the fossil specimen and

that products from the breakdown of proteinaceous material most likely contributed to this organic matter. The peak at 1,033 cm$^{-1}$ presumably is a Si-O stretch mode from microcrystallites of a silicate phase within the fossil. Absorption due to the amide II band, at 1,550 cm$^{-1}$ in the reference spectra, may be present within the broad elevated region of absorbance in the fossil tissue, and may even be slightly shifted to lower wavenumbers and thus contribute to the peak visible at ∼1,510 cm$^{-1}$. At 1,248 cm$^{-1}$ in the fossil specimen, the broad peak would be consistent with amide III but may be convolved with sulfate as seen in the aggrecan spectrum. This region (1,400–1,700 cm$^{-1}$) that includes the amide I and amide II peaks provides the most prominent FTIR absorption peaks of collagen from modern cartilage[26,27]. FTIR mapping shows that the areas of this strong absorbance correlate with those of the

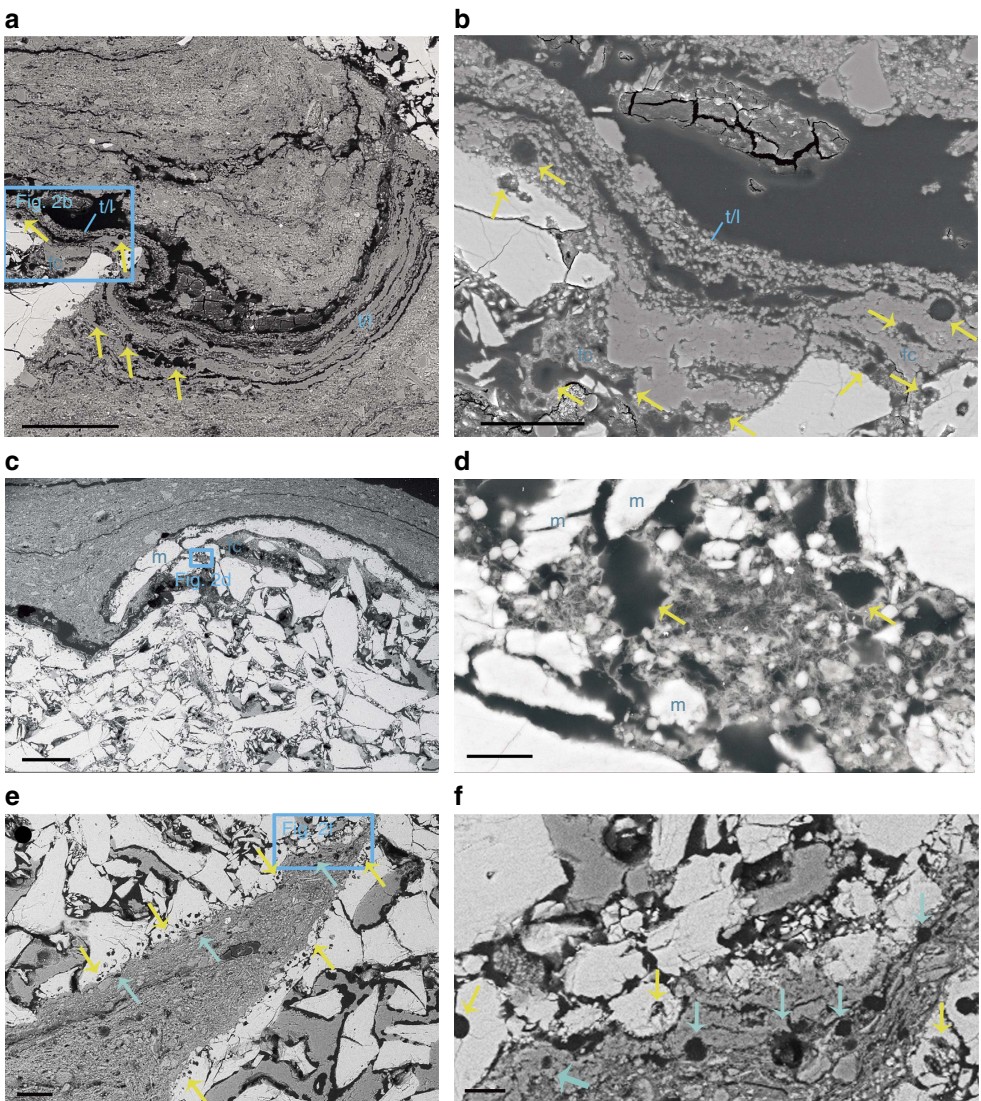

**Figure 2 | Probable tendons/ligaments and fibrocartilages in the *Confuciusornis* ankle.** The intertarsal joint region of the specimen MES-NJU 57002 (Fig. 1f,g). (**a**) BSE-SEM image of the square area in Fig. 1f, showing the tarsometatarsus at the top right of the image and the tibiotarsus on the left, with intervening tendon/ligament material (*t/l*). Within the tendon/ligament, there are regions of fibrocartilage matrix (fc) and cells (arrows) arranged in rows between parallel fibres. (**b**) Close-up view of the square area in **a**. (**c**) BSE-SEM image of the square area in Fig. 1g, showing the tibiotarsus on the left and the tarsometatarsus on the right. A large area of mineralization (m) is visible in the fibrocartilage on the plantar aspect of the ankle joint. (**d**) Close-up view of the square area in **c**. Cellular structures (arrows) are embedded in a matrix composed of network of interwoven fibres and smaller mineralizations. (**e**) BSEM images show that the articular facets of the tarsometatarsus and tibiotarsus in the *Confuciusornis* specimen contain two thin layers of densely packed, paired cellular structures: an inner calcified layer (yellow arrows) and an outer uncalcified layer (blue arrows). (**f**) Close-up view of the square area in **e**. Scale bar, 200 μm in **a** and **c**, 100 μm in **e**, 50 μm in **b**, 10 μm in **d**, 20 μm in **f**.

putative tendons/ligaments and fibrocartilages (areas 2, 3, 5–7 in Fig. 4e,f), although they also overlap the areas of fissures (the black areas in Figs 3a–c and 4e) where the signal probably is interfered with signal from epoxy. These FTIR spectra and mapping imply that amino acid residues may be present.

ToF–SIMS with imaging capability was applied to putative soft tissues on the polished section (the square area in Fig. 4e). Spectrum and image data were acquired in the bunched mode ($m/\Delta m \sim 6{,}000$) at a spatial resolution of $\sim 5\,\mu m$ at analysis depths up to 1 and 1.7 nm, respectively. The spectra presented in Fig. 4b and Supplementary Fig. 7f–j were obtained at analysis depth up to 1 nm from the areas enclosed by the blue line in Supplementary Fig. 7a,b, which mainly consist of the inferred soft tissues. Ion images reveal that the spatial signal intensity distribution for peaks at 275.16, 293.17 and 413.26 AMU in negative ToF–SIMS spectra highly correlates with the distribution of the inferred fibrocartilages (Fig. 4i; Supplementary Fig. 7e,k), and in contrast to those for $NH_4^+$ (Fig. 4j) and $Al^+$ (Fig. 4g) in positive ToF–SIMS spectra, which superimpose with the distribution of fissures that may contain epoxy and the enclosing sediments, respectively. Spatial signal intensity distribution of mass values for peaks at 277.15 AMU (Supplementary Fig. 7c) and 415.26 AMU (Supplementary Fig. 7d) in positive ToF–SIMS spectra also weakly superimposes on the distribution of the inferred soft tissues, but is interfered with by the signal from the sedimentary matrix and probably epoxy in the fissures. Ion images acquired at analysis depth up to 1.7 nm (Supplementary Fig. 7i–t) are consistent with those at analysis depth up to 1 nm.

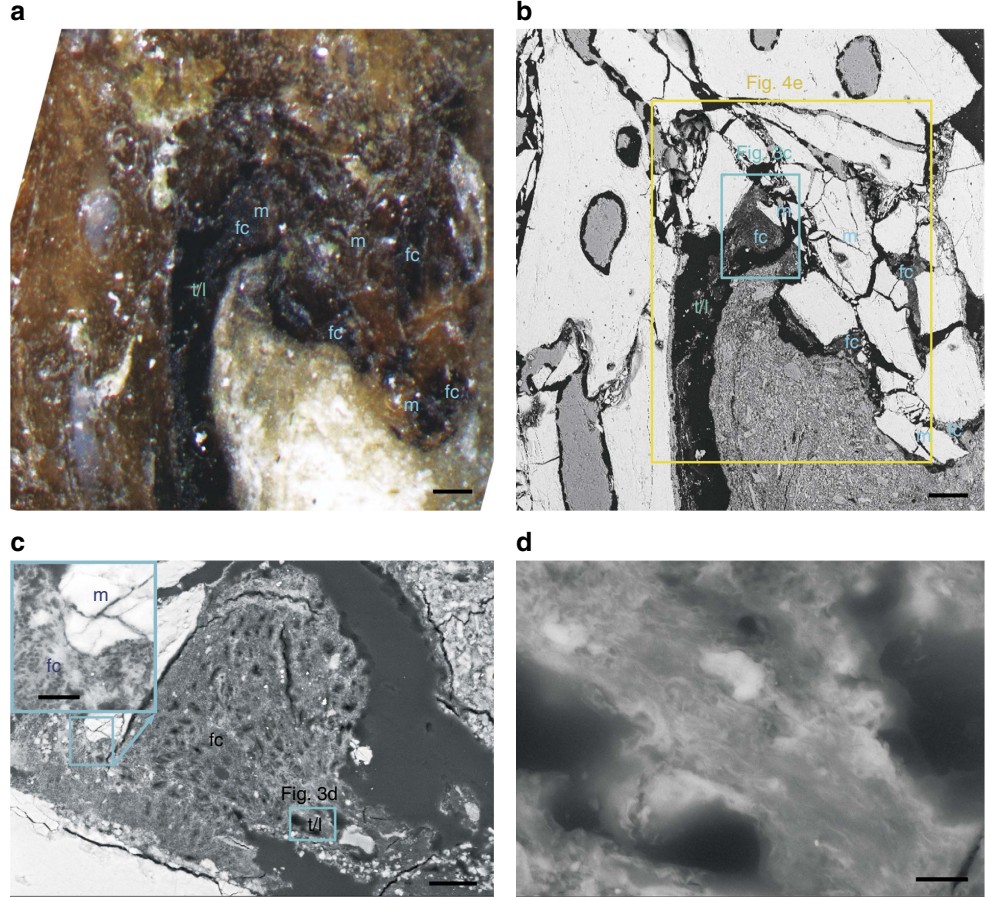

**Figure 3 | Apparent soft tissues in the distal tibiotarsus of the *Confuciusornis* specimen.** Distribution and morphology of the features in specimen MES-NJU 57002 (Fig. 1g). (**a**,**b**) Photographic (**a**) and BSE-SEM image (**b**) show the blackened soft tissues (dark areas) exposed on the plantar surface of the distal tibiotarsus. Note epoxy may be permeated into the fissures (black areas) in the fossil and between the fossil and the enclosing sediments during sample preparation. (**c**) BSE-SEM images of the rectangular area in **b** showing structures of tendons/ligaments and fibrocartilages (insets). (**d**) Close-up view of the tendons/ligaments in **c**. See Figs 1 and 2 for abbreviations. Scale bar, 100 μm in **a** and **b**, 20 μm in **c** and 2 μm in **d**.

This rules out the possibility that spectra resulted from outermost surface contamination.

A characteristic feature of the secondary ion spectrum of an organic molecule M is the appearance of the quasi-molecular ions $(M+H)^+$ $(M^+ cation)^+$, and $(M-H)^-$ (ref. 28). The fragment ions of 275.16 and 293.17 AMU $(=275.17+1H_2O)$ in negative spectrum and 277.15 AMU in positive spectrum probably represent a similar molecule with a mass of 276.16 AMU. Likewise, the measured peaks at 413.26 AMU in negative spectrum and 415.26 AMU in positive spectrum possibly represent a similar molecule with a mass of 414.26 AMU. There are many candidate molecules with the measured ToF–SIMS peak masses and we cannot resolve definitive sequence assignments, because all we have is mass data. Considering that proteins may be degraded into peptides of various sizes in variable states of protonation/deprotonation, the measured mass peaks were tentatively interpreted by comparing probable peptide sequences with the peaks resolved via ToF–SIMS. The 276.16 AMU peptide needed to explain the 275.16 negative peak and 277.16 positive peak could possibly be a Gly-X-Y tripeptide (for example, Gly-Ser-Asn or Gly-Ser-Hyp). The mapped distribution of this mass fragment indicates that it is indeed derived from regions within the fossil that display the presence of amino acids. Because Gly-X-Hyp is one of the most common sequences in collagen, the distribution of this proposed moiety (Fig. 4j; Supplementary Fig. 7) is

consistent with the presence of collagen residue as inferred from the FTIR data. The required peptide mass at 414.26 AMU (to explain the peak at 413.26 AMU in the negative spectrum and 415.26 AMU in the positive spectrum) is more problematic. Although it could represent protein-derived residue, given its distribution we do not base any conclusions on this resolved mass. However, we conclude that the ∼276 AMU data are consistent with the presence of Gly-X-Y residues. Therefore, the ToF–SIMS maps and spectra strengthen the inference from the FTIR spectra that amino acid residues from collagen may be present in the fibrocartilage zones of this specimen.

**X-ray chemical analyses.** Connective tissues such as cartilage or tendons/ligaments normally have high-sulfur content and their dominant sulfur species is sulfate. Synchrotron rapid-scanning X-ray fluorescence mapping revealed that the sulfur content of the soft tissues was higher than the overlying bone, the embedding sediments and the fissures between them (Fig. 4d). Subsequent X-ray absorption near-edge spectroscopy showed that the sulfur speciation in this region of the fossil is dominated by sulfate (Fig. 4c). This is wholly consistent with the FTIR peak at 1,248 cm$^{-1}$, which relates to an overlapping absorbance region of Amide III from collagen and sulfate from aggrecan. Since the region of high sulfur was confined to the soft tissues, the sulfur

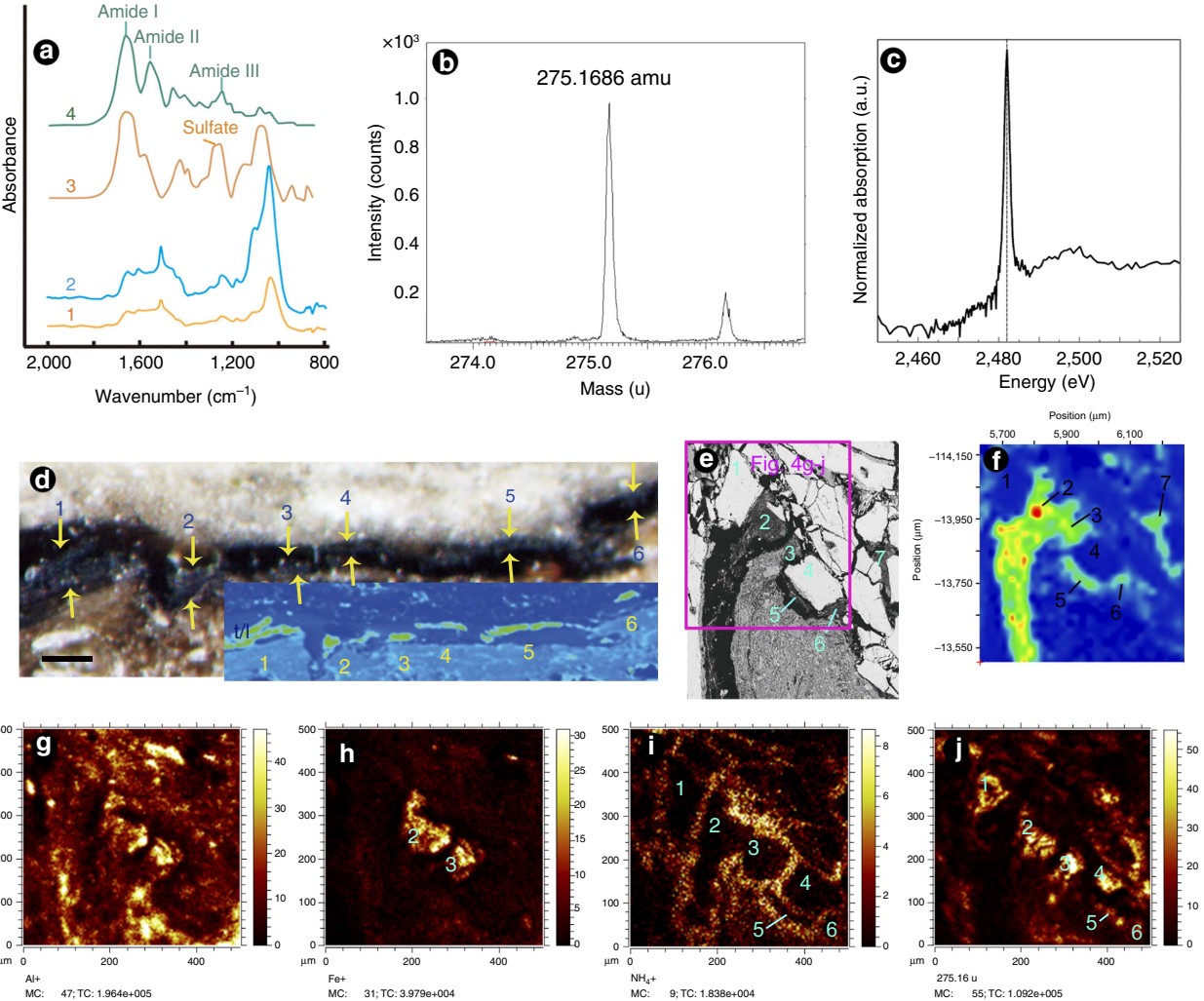

**Figure 4 | Chemical analyses of the putative soft tissues in the *Confuciusornis* hindlimb.** (**a**) Comparison of FTIR spectra (1–2) obtained from two spots on the supposed soft tissues with the typical proteoglycan aggrecan (3) and collagen (4)[26]. (**b**) Mass spectra of detailed region in 274–276.5 AMU of the supposed soft tissues. (**c**) Sulfur K-edge XANES spectrum from the high-sulfur region showing that the dominant sulfur species is sulfate (edge position for sulfate standard at 2,482 eV indicated by vertical dashed line). (**d**) False colour synchrotron microfocus XRF map of sulfur in the rectangular area in Fig. 1g shows that the high-sulfur regions (yellow-green colour, inset:#1–6) are correlated with the distribution of tendons/ligaments (#1–6). Scale bar, 100 μm. (**e–j**) Comparison of the distribution of the inferred fibrocartilages (#2–3, 5–7) and possibly mineralized fibrocartilages (#1, 4) in BSE-SEM image of the rectangular area in Fig. 3b (**e**) and FTIR map of the strong absorbance region of 1,400–1,700 cm$^{-1}$ in the same area (**f**) and ion images of the spatial signal intensity distribution in the square area in Fig. 4e for Al$^{+}$ (**g**), Fe$^{+}$ (**h**), NH$_4^{+}$ (**i**) and the peaks at 275.16 AMU (**j**) in negative ToF–SIMS spectra. The FTIR map was collected using a Thermo IN10MX infrared microscope with a cooled MCT detector.

may derive from original sulfate or the breakdown of original organosulfur compounds.

**Summary of chemical analyses.** These chemical analyses support our inference that the segmented biological structures probably are the preserved residues of tendons/ligaments and cartilages. Interestingly, an ion image of spatial signal intensity distribution for Fe$^{+}$ (Fig. 4h) in the positive ToF–SIMS spectra can also be directly superimposed onto the inferred distribution of soft tissues, which supports the hypothesis that iron may be involved in the preservation of soft tissue[5,11].

**Morphological interpretations of likely soft tissues.** The deepest tendons that cross the plantar surface of the lower leg in extant birds are parts of the digital flexor muscles[4,29,30]. This implies that the preserved residual tendons/ligaments are remnants of those digital flexor tendons, and intertarsal

ligaments closely associated with them. The tendons/ligaments pass through two cartilaginous structures, a sulcus on the distal tibiotarsus and a ridge on the proximal end of the tarsometatarsus.

The sulcus is delimited by two cristae against the condyles on the distal tibiotarsus. The cristae, about 1 mm thick, are porous and locally fragmented. Their lower parts are fused to the underlying bone, and their upper surfaces are irregular and covered by partly mineralized fibrocartilages in the area contacting with tendons/ligaments (Figs 1e–l and 3a,b). Partly intact preservation of spongy condyles and continuous cortical bone across the cristae and sulcus (Fig. 1i–l) indicate that the cristae are not derived from compression or displaced cortical bone, even though the bone apparently underwent some compression during fossilization. The fibrocartilaginous nature of the cristae is supported by the distinct light grey colour of the cristae in comparison with the brown colour of the remaining bone exposed in a further two *Confuciusornis*

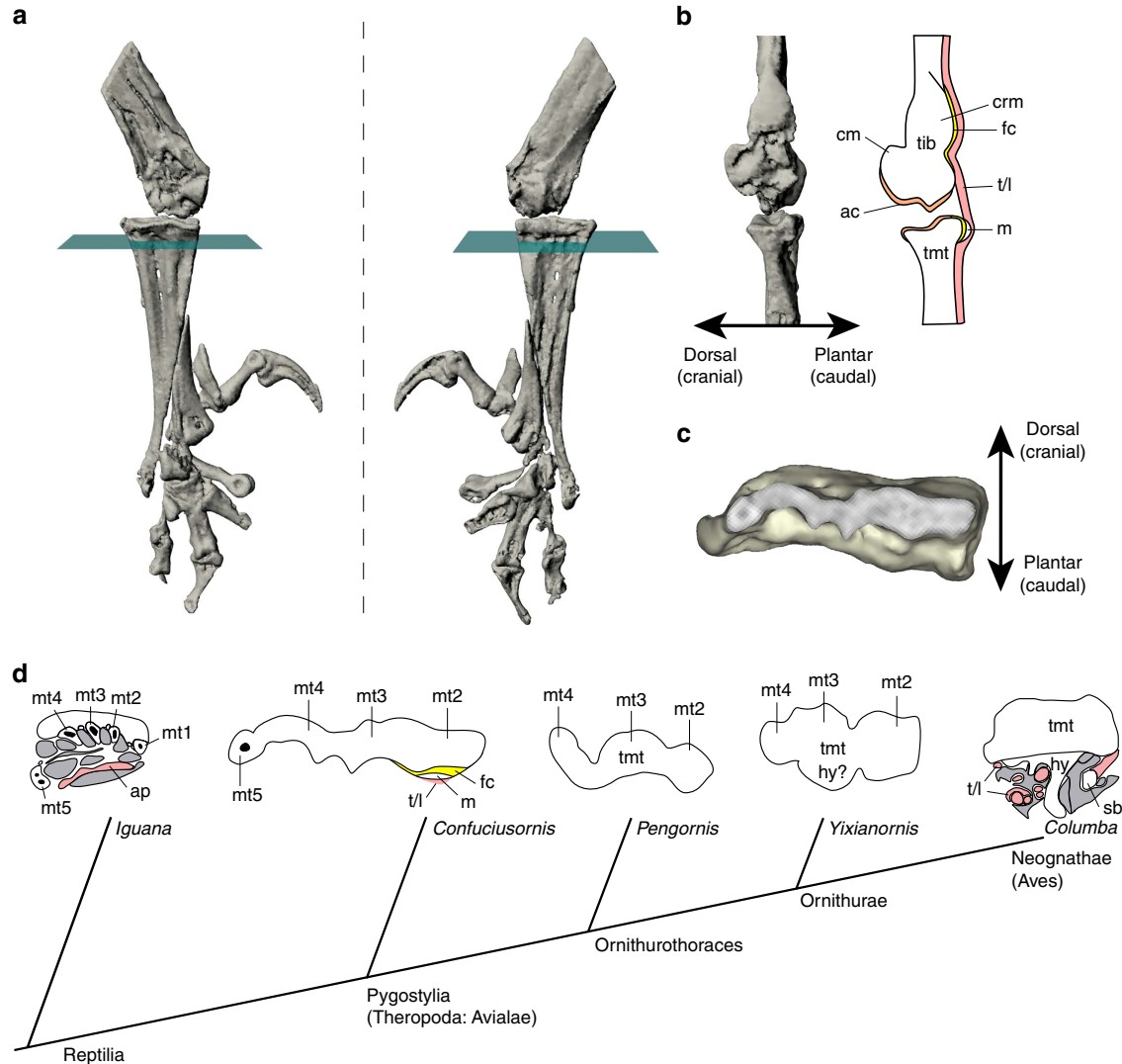

**Figure 5 | Evolution of ankle anatomy in early birds.** (**a**) Dorsal/cranial (left) and plantar/caudal (right) views of the right pes (and, more obliquely and proximally, the tibia) of *Confuciusornis* using scan data from Fig. 1, showing where virtual sections (blue-green planes perpendicular to bones) were taken for use in **b**–**d**. (**b**) Inferred distribution of the preserved tendons/ligaments, fibrocartilages, articular cartilages, cartilaginous cristae and ridges in the right ankle joint of *Confuciusornis*. Medial view emphasizing the wrap-around tendons of the digital flexors, associated fibrocartilage and mineralization. (**c**) Cross-sectional reconstructed scan data from the proximal tarsometatarsus as shown in **a** and used in **d**. (**d**) Drawing of histological horizontal cross-sections through the proximal end of the tarsometatarsus (or metatarsus) of squamate *Iguana*[38], pygostylian bird *Confuciusornis*, enantiornithine bird *Pengornis*, ornithuran bird *Yixianornis* and crown-group bird *Columba*[39] shows our hypothesis for the evolution of the hypotarsus (hy) and other derived features of avian ankles. Pink colours are tendinous/ligamentous tissue; white is bone; grey is muscular tissue and yellow is (fossilized) fibrocartilage. From left to right, as the number of main toes was reduced and the pes narrowed and consolidated, becoming more robust, it became more plantarly projected (that is, a larger hypotarsus) and formed fibrocartilages, mineralizations and ultimately bony ridges and grooves to enhance the leverage of as well as enclose and guide the pedal tendons. See Figs 1 and 2 for abbreviations. Not to scale. Virtual 'histological' sections for the simple sketches of *Pengornis* (IVPP V15336) and *Yixianornis* (IVPP V12558) were done digitally using microCT scan data from ref. 3, at the approximate same proximodistal position as the other images, following the procedure shown in **a** and **c**.

specimens from the Institute of Vertebrate Paleontology and Paleoanthropology, Chinese Academy of Sciences (IVPP; IVPP 18156 and IVPP 13175; Supplementary Fig. 6). The sulcus and the contacting fibrocartilage anatomically correlate with the caudal intercondylar sulcus and the tibial cartilage that is situated in the sulcus in extant birds[29,30].

The ridge on the proximal end of the tarsometatarsus is composed of fibrocartilage and a crescent-shaped mineralization ∼0.1 mm thick on its plantar side (Figs 1g,2c,d and 5). It is elevated and inflated in its medial to central position. Distal to the ridges, sulci are formed on the central to lateral position of the proximal tarsometatarsus, which connect to the grooves

against metatarsals II, III and IV in the area where the metatarsals are not fused (Fig. 1m–p). Such a cartilaginous ridge with sulci distal to it can be observed in other *Confuciusornis* specimens (for example, IVPP18156 and IVPP 18168; Supplementary Fig. 6). The ridge occurs at the position of the hypotarsus in ornithuromorph birds[31], but it is less distinct and more cartilaginous compared with the latter structure, even with those seen in early ornithuromorph birds, such as the Late Cretaceous *Patagopteryx*, *Pengornis*, *Yixianornis*, *Apsaravis* and *Ichthyornis*, which have a flat bony projection or unprojected discrete surface without canals and sulci[32–34]. In contrast, the ridge resembles an intermediate state of the hypotarsus in the

ontogeny of extant birds, in which the hypotarsus remains cartilaginous until the latest stages or after hatching, when it ossifies from a separate centre located on its distal medial corner[35,36].

## Discussion

Overall, this specimen of *Confuciusornis* reveals the remarkable preservation of residual tendons/ligaments belonging to or associated with the digital flexor muscles passing through two musculoskeletal structures comparable to two novel structures in the lower legs of crown-group birds: the tibial cartilage and hypotarsus (Fig. 5). The weaker, more cartilaginous expression of the two structures in *Confuciusornis* compared to many extant birds indicates that they might be either homoplastic or intermediate character states in the evolution of the highly derived musculoskeletal apparatus that characterizes crown-group birds. It may seem implausible that these specializations could have been overlooked in so many other fossil birds (for example, thousands of known *Confuciusornis* specimens), but they are small and subtly expressed, like those in the late ontogeny of extant birds.

The evolution of an incipient tibial cartilage and hypotarsus has important implications for the evolution of hindlimb function in birds. The digital flexor tendons pass through specific perforations, sulci or canals in these two structures to their insertions on the plantar surfaces of the distal ends of the digits in extant birds[29,30]. The orientation of these structures enhances not only the moment arms (leverages), but also the control of the muscles that flex and manipulate the corresponding digits[4,37]. The origin and early evolution of the two guiding structures is unclear. We infer that there was a transition between the ancestral state in reptiles, in which digital flexor tendons contribute to the complex plantar aponeurosis[4,38], and an intermediate state in early ornithuromorph birds, involving an incipient osseous hypotarsus and tibial cartilage as interpreted here (Fig. 5)[4,31]. The largely cartilaginous nature of these two structures in *Confuciusornis* hints that their derived state in crown-group birds may have evolved from the gradual fusion of mineralized cartilage, similar to a traction epiphysis—a pre-existing intratendinous ('metaplastic') mineralization that has become secondarily fused to the main bone[39].

Fibrocartilage tends to develop within tendons/ligaments where they experience compression[20,40]. It is a dynamic tissue that can be maintained *in vitro* when tendon is compressed, and disappears when the strain is removed. Fibrocartilaginous wrap-around tendons commonly occur in areas where they are strongly bent around their pulleys and heavily loaded (such as in the human ankle), in contrast to areas where they only change direction when the limb is in certain positions (such as the human wrist) in which non-fibrocartilaginous tendons tend to develop[20]. Thus, the occurrence of the thick fibrocartilage and cartilaginous structures on the plantar surface of the ankle joint of *Confuciusornis* implies that the tendons of digital flexors may have frequently experienced high compressive loading from being habitually wrapped around the ankle joint. These tendons function mainly in flexing the digits, although they contribute to extending the ankle joint late in the stance (supportive) phase of locomotion[29,30]. Their relatively derived position that we infer here for *Confuciusornis* and later birds supports the inference that a more crouched limb position, and overall hindlimb function that was intermediate between the more vertically-oriented limbs of early dinosaurs and the more crouched limbs of extant birds, had evolved by this point in the evolution of birds, which is consilient with other evidence[1–3]. The results from the chemical analyses allow us to conclude that organic residue is correlated with the structures we have resolved. Our data are not sufficient to definitively resolve the source of this residue; however the analytical results are consistent with these organics being derived from a collagen precursor, thus inductively strengthening our morphological interpretation. It is only via the new combination of techniques from molecular palaeontology and imaging analysis with classic approaches from comparative anatomy, palaeontology and biomechanics that we have discovered such remarkably fine details of fossilized soft tissue preservation in *Confuciusornis* and their significance for the evolution of locomotor adaptations in the bird lineage.

## Methods

**Fossil preparation and imaging.** The unprepared *Confuciusornis* specimen MES-NJU 57002 was identified by microCT scanning using a 2010 GE phoenix v|tome|x s240 system. MicroCT scan data are available[41]. The specimen was compared with others (Supplementary Table 1) to confirm its identity and adult status.

The specimen was strengthened by epoxy. Polished cross-sections and thin-sections were prepared for observation and then imaged using a polarized-light microscope and scanning electron microscope equipped with back-scattered electron (BSE) detector and EDX detector. SEM analysis used LEO1530 VP, which is an environmental s.e.m. capable of operating in variable pressure mode. Additional s.e.m. image data are available[42].

**FTIR microscopy.** The FTIR map was collected using a Thermo IN10MX infrared microscope with a cooled MCT detector. Each spectrum was collected from an apertured area of $30 \times 30\ \mu$ at the sample, with a $20\ \mu$ step in the $x$ and $y$ directions. The spectra resolution was $8\,cm^{-1}$ and 16 scans were collected at each point. A Kramers–Kronig transformation (applied within the Picta software) was found to give the best absorption line shapes and was therefore applied to each spectrum. FTIR data are available[43].

**ToF–SIMS.** ToF–SIMS analysis was carried out using a ToF–SIMS V instrument (IONTOF GmbH) at the Analysis Center of Tsinghua University. ToF–SIMS analyses in the static SIMS mode were performed using 30 keV $Bi^{3+}$ primary ions and low-energy electron flooding for charge compensation. Spectrum and image data were acquired in the bunched mode ($m/\Delta m \sim 6{,}000$) at a spatial resolution of $\sim 5\,\mu m$ at $128 \times 128$ pixels. ToF-SIMS data are available[44].

**Synchrotron X-ray fluorescence and absorption spectroscopy.** Synchrotron X-ray fluorescence (XRF) elemental imaging and X-ray absorption spectroscopy (XAS) was carried out at Diamond Light Source (DLS, Oxford, UK) microfocus beamline I18 using Kirkpatrick–Baez mirrors to produce a spot size of approximately 5 μm, a double crystal Si (111) monochromator to scan incident beam energy and a 4-element Vortex silicon drift detector. Flux was estimated to be between $10^8$ and $10^9$ photons s$^{-1}$. XRF maps were produced using the PyMCA freeware[45] ROI imaging tool by defining the X-ray emission energy of an element in the recorded spectra and displaying the intensity of X-ray counts for that element across the mapped area. A $ZnSO_4$ standard was used to calibrate the sulfate absorption edge energy in the XAS experiments. XRF[46] and XAS[47] data are available.

**Comparative anatomy.** The right ankle of an adult helmeted guineafowl (*Numida meleagris*) was dissected (Supplementary Fig. 4) to supplement the literature on this region's morphology in birds. For comparative microscopic anatomy, the left ankle of an adult quail (*Coturnix coturnix*) was also studied histologically (images of the guineafowl showed similar anatomy but are not included here). The quail ankle was decalcified in 5% formic acid and cut at an oblique angle similar to the fossil plane, wax embedded, then sections were cut at 4 μ and stained with standard haematoxylin and eosin for illustrating comparative bone and cartilage tissue anatomy (Supplementary Fig. 5). Cadavers were donated from local breeders for research purposes and had died for unrelated reasons.

**Data availability.** The data that support the findings of this study are available in figshare with the identifier(s) as follows: microCT scan data[41] (doi: 10.6084/m9.figshare.4595971), additional s.e.m. images[42] (doi: 10.6084/m9.figshare.4595929), FTIR data[43] (doi:10.6084/m9.figshare.4595920), ToF–SIMS data[44] (doi: 10.6084/m9.figshare.4595932), XRF mapping data[46] (doi: 10.6084/m9.figshare.4595941) and XAS data[47] (doi: 10.6084/m9.figshare.4595938).

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

## Acknowledgements

We thank Zhonghe Zhou, Jin Meng, Junfeng Ji, Xiancai Lu, Julia Clarke and Maria McNamara for discussions, and Yan Fang, Wuping Li, Tong He, H. Jones, M. Hethke, M. Hill, A. Davidson, Junying Ding and Huan Liu for laboratory assistance. This work was supported by the National Science Foundation of China (41672010, 41688103) and State Key Laboratory of Palaeobiology and Stratigraphy (Nanjing Institute of Geology and Palaeontology, Chinese Academy of Sciences) (153104) to B.J., as well as a Royal Society Leverhulme Trust senior research fellowship, Leverhulme Trust research Grant Number RPG-2013-108 and Natural Environment Research Council Grant Number NE/K004751/1 to J.R.H. We thank the staff at the Diamond Light Source, beamline I18 (beam allocation SP9488). We also thank Nick Lockyer for discussions about the ToF–SIMS data.

## Author contributions

B.J. and J.R.H. designed the research, B.J. and T.Z. studied the histology (including SEM) of the fossils, S.R. and J.R.H. made the histological comparisons between fossils and extant birds, N.P.E. and R.A.W. carried out the synchrotron rapid-scanning X-ray fluorescence and X-ray absorption near-edge structure spectroscopies, B.J. did the ToF–SIMS analysis, M.J.B. and S.C.K. did the FTIR analysis, Z.L. and S.R. did the microCT reconstructions, and B.J., R.A.W., M.J.B. and J.R.H. wrote the paper; all authors approved the final draft of the paper.

## Additional information

**Competing interests:** The authors declare no competing financial interests.

