## [Peer Review File · Nature Communications]

Reviewers' comments:

Reviewer #1 (Remarks to the Author):

In the manuscript entitled "Cellular preservation of musculoskeletal specializations in the early bird *Confuciusornis* revealed by integration of imaging and chemical analyses" the authors describe fossil evidence of anatomical modification in the lower hindlimb of theropod dinosaurs changed appreciably on the line to extant birds. My expertise in molecular analysis of ancient protein residues and I will limit my comments to this part of the manuscript.

The authors claim they detected traces of ancient organic material by chemical analyses. In particular they provide evidence supporting the identification of protein residues. In my opinion this claim is not supported strongly enough. The FTIR data presented in fig. 3g are, at best, barely compatible, or not in contrast, with the identification of protein residues, but it does not represent, in my opinion, a strong piece of evidence unambiguously supporting this interpretation.

Without further elaboration, the statement: "FTIR mapping shows that the area of this strong absorbance is highly correlated with that of the putative tendons/ligaments and fibrocartilages (Fig. 3h).", sounds quite arbitrary.

It cannot be excluded that traces of organic residues can actually be observed in the samples described, in agreement with the anatomical reconstruction the authors present. However, the evidence provided is not sufficient to identify them as protein residues, and as a consequence not sufficient to identify them as the residues of specific proteins: collagen and aggrecan in this case.

The case would be reinforced by providing some molecular evidence, and not just spectroscopic data, compatible with protein degradation products, possibly by using TOF-SIMS analysis, as described in:

Biomolecular Characterization and Protein Sequences of the Campanian Hadrosaur *B. canadensis*

Mary H. Schweitzer, et al.

Science 324, 626 (2009);

DOI: 10.1126/science.1165069

Nat Commun. 2012 May 8;3:824. doi: 10.1038/ncomms1819.

Molecular preservation of the pigment melanin in fossil melanosomes.

Lindgren J1, Uvdal P, Sjövall P, Nilsson DE, Engdahl A, Schultz BP, Thiel V.

Reviewer #2 (Remarks to the Author):

The authors present a fascinating application of imaging technology with interpreting the

anatomy of the hindlimb joints of a fossil bird. The inferences born by spectroscopy and other imaging tools apparently can distinguish different types of soft tissues based on morphology and then the functional and evolutionary significance of the new soft tissue data can be interpreted, in this case the characteristic ankle/hypotarsal joints and tendons of bird hindlimbs. This is cool and helps illustrate how many more data can be pulled from fossils such as the slab fossils from China.

I don't have the expertise to review the raw data analysis from the non-CT imaging tools, thus I must assume the 'well-documented' approach, though supported by only a single citation by Manning, is validation enough of these approaches.

The anatomical and functional inferences appear robust given the data. Unfortunately, Figure 4 which illustrates the anatomy and function (and evolution?) of the system is hardly of publishable quality. The ankle in A is not in the same position or posture in the drawings. Perhaps different renders of the CT data can be used to better illustrate the morphology? The line drawing (to use the term generously) of the Patagopteryx cross section offers little to readers and the use of color in the figure seems inconsistent. Shouldn't the tendons in the Iguana cross section also be pink? Following on the heels of the good 3D renders and illustrations of fossils, in general Figure 4 is not acceptable and is rather an embarrassing contribution for a functional morphology paper.

Reviewer #3 (Remarks to the Author):

A/ Summary of the key results

Microscopic investigations of bone thin sections of a cretaceous early bird fossils show the exceptional preservation of cellular structures in soft tissue. FTIR and absorption spectroscopy suggested that cellular structures are associated with preserved organic matter. The combination of imaging and chemical analysis provided unique information about first steps of anatomical and histological modifications involved in evolution of bipedal gait of bird.

B Originality and interest: if not novel, please give references

The fossil specimen studied present soft tissues particularly well preserved. EDX and FTIR analyzes suggest the presence of carbon and organic matter preserved in these areas. This represents a rare case of exceptional preservation. But beyond this evidences of preservation of biological structure over geological timescales, researchers are developing an innovative approach combining paleontological information, histological and chemical. This original study allows them to express hypotheses on the origin and the biological mechanisms of appearance of bipedal gait of birds.

C Data & methodology: validity of approach, quality of data, quality of presentation

The approach combining paleontology, histology and physicochemical characterization is well developed and appropriate to the study. Data are well exposed even if quality of

illustrations could be enhanced.

E.g.: the figure 3 contains too much information and it is difficult to understand. The methods used for mapping must be indicated in Figures 3b and 3h. FTIR results presented both in figures of 3d and 3h must be presented together on a separate figure.

Details of the method must be addressed:

L 381 - The procedure used for the preparation of thin section is not very clear. Is a embedding resin was used? It seems not, but this must be stated (if a resin has been used, a FTIR spectrum must be presented)

L381 - FTIR spectra were treated using Kramers Kroning suggesting the analysis of polished section in reflection mode. Acquisition mode has to be mentioned in the manuscript.

D Appropriate use of statistics and treatment of uncertainties

-

E Conclusions: robustness, validity, reliability

Microscopic observations and the inferences based on morphological similarities with modern soft tissues are convincing.

Some information obtained by physical analysis seems less robust and could be qualified:

For example Figure 3g shows a FTIR spectrum of soft tissue, compared to the spectrum of collagen and proteoglycan aggrecan. Spectrum 1 (soft tissue) shows bands around 1650 and 1248 cm^{-1} . However, the intensity ratios are very different compared to modern references and it is difficult to clearly identify Amide II band. In my opinion, it seems difficult to conclude definitively for the presence of aggrecan and collagen, even if the presence of organic matter is suggested. Furthermore, only one spectra of soft tissue is presented. It could be interesting to present several spectra obtained in order to assess the variability (and consistency) of composition in this area. Anyway, the clear identification of preserved organic matter is not critical for the publication because the microscopic observations are sufficient and does not affect the conclusion of this study.

F Suggested improvements: experiments, data for possible revision

G References: appropriate credit to previous work?

-

H Clarity and context: lucidity of abstract/summary, appropriateness of abstract, introduction and conclusions

Methods used to achieve results must be indicated in the abstract.

Response to Reviewers

Reviewers' comments: (and our Responses in bold font)

Reviewer #1 (Remarks to the Author):

In the manuscript entitled "Cellular preservation of musculoskeletal specializations in the early bird *Confuciusornis* revealed by integration of imaging and chemical analyses" authors describe fossil evidence of anatomical modification in the lower hindlimb of theropod dinosaurs changed appreciably on the line to extant birds. My expertise in molecular analysis of ancient protein residues and I will limit my comments to this part of the manuscript.

The authors claim they detected traces of ancient organic material by chemical analyses. In particular they provide evidence supporting the identification of protein residues. In my opinion this claim is not supported strongly enough. The FTIR data presented in fig. 3g are, at best, barely compatible, or not in contrast, with the identification of protein residues, but it does not represent, in my opinion, a strong piece of evidence unambiguously supporting this interpretation.

Without further elaboration, the statement: "FTIR mapping shows that the area of this strong absorbance is highly correlated with that of the putative tendons/ligaments and fibrocartilages (Fig. 3h).", sounds quite arbitrary.

It cannot be excluded that traces of organic residues can actually be observed in the samples described, in agreement with the anatomical reconstruction the authors present. However, the evidence provided is not sufficient to identify them as protein residues, and as a consequence not sufficient to identify them as the residues of specific proteins: collagen and aggrecan in this case.

--We agree that the statement "FTIR mapping shows that the area of this strong absorbance is highly correlated with that of the putative tendons/ligaments and fibrocartilages (Fig. 3h)." seemed arbitrary. We thus modified this part to (lines 100-105) "FTIR mapping shows that the areas of this strong absorbance correlate with those of the putative tendons/ligaments and fibrocartilages (areas 2, 3, 5-7 in Fig. 4e-f), although they also overlap the areas of fissures (the black areas in Fig. 3a-c and Fig. 4e) where the signal probably is interfered with signal from epoxy. These FTIR spectra and mapping imply that amino acid residues may be present".

The case would be reinforced by providing some molecular evidence, and not just spectroscopic data, compatible with protein degradation products, possibly by using TOF-SIMS analysis, as described in:

Biomolecular Characterization and Protein Sequences of the Campanian Hadrosaur *B. canadensis*
Mary H. Schweitzer, et al.
Science 324, 626 (2009);
DOI: 10.1126/science.1165069

Nat Commun. 2012 May 8;3:824. doi: 10.1038/ncomms1819.
Molecular preservation of the pigment melanin in fossil melanosomes.
Lindgren J1, Uvdal P, Sjövall P, Nilsson DE, Engdahl A, Schultz BP, Thiel V.

--The comments and suggestions are very constructive. We carefully read the recommended references and carried out ToF-SIMS analysis on the studied sample. The results were added into the text (Lines 106-136), further supporting our inference.

Reviewer #2 (Remarks to the Author):

The authors present a fascinating application of imaging technology with interpreting the anatomy of the hindlimb joints of a fossil bird. The inferences born by spectroscopy and other imaging tools apparently can distinguish different types of soft tissues based on morphology and then the functional and evolutionary significance of the new soft tissue data can be interpreted, in this case the characteristic ankle/hypotarsal joints and tendons of bird hindlimbs. This is cool and helps illustrate how many more data can be pulled from fossils such as the slab fossils from China.

I don't have the expertise to review the raw data analysis from the non-CT imaging tools, thus I must assume the 'well-documented' approach, though supported by only a single citation by Manning, is validation enough of these approaches.

--We agree with the referee here, and modified the text by adding more references (#13-19) to the chemical analysis section.

The anatomical and functional inferences appear robust given the data. Unfortunately, Figure 4 which illustrates the anatomy and function (and evolution?) of the system is hardly of publishable quality. The ankle in A is not in the same position or posture in the drawings. Perhaps different renders of the CT data can be used to better illustrate the morphology? The line drawing (to use the term generously) of the Patagopteryx cross section offers little to readers and the use of color in the figure seems inconsistent. Shouldn't the tendons in the Iguana cross section also be pink? Following on the heels of the good 3D renders and illustrations of fossils, in general Figure 4 is not acceptable and is rather an embarrassing contribution for a functional morphology paper.

--We have put substantial effort into revising Figure 4 (now 5), emphasizing the morphology of the specimen as recommended. We have also changed the colour to be more consistent in the bottom row, and added some illustrations of other ankle regions (relying on actual microCT data we have from other fossil birds, as cited; rather than more ambiguous literature sketches) to improve the clarity of our depiction of the evolutionary changes of morphology across the lineage shown. The most fundamental problem we have faced in this figure is that, even in the best specimens and

with microCT scan data, preservation does not easily enable illustration of the small or subtle morphological details such as those described here. What the illustrations of *Pengornis* and *Yixianornis* that we have added (and thereby removing *Patagopteryx*, which was more speculative) show is the gross cross-sectional shape of the proximal tarsometatarsus, and hints of the incipient hypotarsus that the existing literature already ascribes to them (but not to *Confuciusornis*, which is this study's major focus and contribution). We hope that the combination of detailed, high-resolution ~3D skeletal images in Figs 1,5a-c and the extended data with the simplified, more abstract sketches in Fig 5d get our main points across in ways that a general audience and morphologists will now appreciate more.

Reviewer #3 (Remarks to the Author):

A/ Summary of the key results

Microscopic investigations of bone thin sections of a cretaceous early bird fossils show the exceptional preservation of cellular structures in soft tissue. FTIR and absorption spectroscopy suggested that cellular structures are associated with preserved organic matter. The combination of imaging and chemical analysis provided unique information about first steps of anatomical and histological modifications involved in evolution of bipedal gait of bird.

B Originality and interest: if not novel, please give references

The fossil specimen studied present soft tissues particularly well preserved. EDX and FTIR analyzes suggest the presence of carbon and organic matter preserved in these areas. This represents a rare case of exceptional preservation. But beyond this evidences of preservation of biological structure over geological timescales, researchers are developing an innovative approach combining paleontological information, histological and chemical. This original study allows them to express hypotheses on the origin and the biological mechanisms of appearance of bipedal gait of birds.

C Data & methodology: validity of approach, quality of data, quality of presentation

The approach combining paleontology, histology and physicochemical characterization is well developed and appropriate to the study. Data are well exposed even if quality of illustrations could be enhanced.

E.g.: the figure 3 contains too much information and it is difficult to understand. The methods used for mapping must be indicated in Figures 3b and 3h. FTIR results presented both in figures of 3d and 3h must be presented together on a separate figure.

-- We tried to enhance the quality of all the illustrations and separated the original Figure 3 into two figures: a morphological one and another chemical one (now Figs. 3,4). The methods used for mapping were indicated in the new Figure 4b and 4f.

Details of the method must be addressed:

L 381 - The procedure used for the preparation of thin section is not very clear. Is a embedding resin was used? It seems not, but this must be stated (if a resin has been used, a FTIR spectrum must be presented)

--Yes, the sample was consolidated by epoxy and we have clarified this in the text (e.g. lines 104,116,120,382/Fig3 caption,423/Methods). Since we carried out ToF-SIMS spectroscopy with

mapping capability, the result helps to exclude the interference of the epoxy (see the new Figure 4g-j). We did not provide a FTIR spectrum for the epoxy here.

L381 - FTIR spectra were treated using Kramers Kroning suggesting the analysis of polished section in reflection mode. Acquisition mode has to be mentioned in the manuscript.

--Yes, we added this (lines 431-433).

D Appropriate use of statistics and treatment of uncertainties

-

E Conclusions: robustness, validity, reliability

Microscopic observations and the inferences based on morphological similarities with modern soft tissues are convincing.

Some information obtained by physical analysis seems less robust and could be qualified:

For example Figure 3g shows a FTIR spectrum of soft tissue, compared to the spectrum of collagen and proteoglycan aggrecan. Spectrum 1 (soft tissue) shows bands around 1650 and 1248 cm⁻¹.

However, the intensity ratios are very different compared to modern references and it is difficult to clearly identify Amide II band. In my opinion, it seems difficult to conclude definitively for the presence of aggrecan and collagen, even if the presence of organic matter is suggested.

Furthermore, only one spectrum of soft tissue is presented. It could be interesting to present several spectra obtained in order to assess the variability (and consistency) of composition in this area.

Anyway, the clear identification of preserved organic matter is not critical for the publication because the microscopic observations are sufficient and does not affect the conclusion of this study.

--We obtained several FTIR spectra on the soft tissues. All the spectra seem similar and we choose the two best (most representative) ones to represent them; it would be superfluous to present all of them. In addition, we carried out ToF-SIMS spectroscopy with mapping capability (see response to reviewer 1)-- those results help a lot to distinguish the preserved organic matter from the epoxy and sediments (see the new Figure 4g-j; also comments above/new text on epoxy).

F Suggested improvements: experiments, data for possible revision

G References: appropriate credit to previous work?

-

H Clarity and context: lucidity of abstract/summary, appropriateness of abstract, introduction and conclusions

Methods used to achieve results must be indicated in the abstract.

--We cited as many references as we felt necessary to support our points, and tried to spread them out across topics/authors. Due to the very tight limit of abstract length, we could not add there the methods used to achieve our results.

REVIEWERS' COMMENTS:

Reviewer #1 (Remarks to the Author):

I carefully read the authors' reply to my comments and, although I appreciate they accepted my suggestion to include ToF-SIMS analysis, I also have to say that some statements in the section describing the results of molecular analysis still look inconclusive to me. Specifically:

Lines 93-97:

"The peak at 1033 cm⁻¹ presumably is a Si-O stretch mode from microcrystallites of a silicate phase within the fossil. Absorption due to the amide II band, at 1550 cm⁻¹ in the reference spectra, may be present within the broad elevated region of absorbance in the fossil tissue, and may even be slightly shifted to lower wavenumbers and thus contribute to the peak visible at ~1510 cm⁻¹."

Arguments phrased using expressions like: "presumably", "may be present" and "may even be slightly shifted" look more speculative than conclusive to me. They look to me more based on personal interpretation rather than strong evidence.

Also the interpretation of the ToF-SIMS results is tentative. In lines 131-135 the authors claim: "The peak at 275.16 amu in the negative spectrum is approximately consistent with a tetrapeptide -Ser-Gly-Gly-Gly- with a mass of 276.13 amu ($C_3H_7NO_3 + 3C_2H_5NO_2 - 3H_2O = C_9H_{16}N_4O_6$), while the peak at 413.26 amu in the negative spectrum accords roughly to a tetrapeptide -Thr-Thr-Pro-Pro- with a mass of 414.24 amu ($2C_4H_9NO_3 + 2C_5H_9NO_2 - 3H_2O = C_{18}H_{30}N_4O_7$)."

The identification of short peptides by "approximately" or "roughly" matching them to ToF-SIMS masses doesn't seem rigorous enough to me. If we then look at the suggested sequences: Ser-Gly-Gly-Gly and Thr-Thr-Pro-Pro we don't see the G-X-Y pattern typical of collagen: the most abundant proteins originally present in the fossilised tissue and arguably the one that has higher chances to be recovered. In the authors opinion, to which proteins do the identified peptide sequences belong to?

In conclusion in my opinion the authors should add to their text a clear cautionary note indicating that all their interpretation of the molecular results are tentative and compatible with, but not conclusively proving, the preservation of ancient biomolecules.

Response to Reviewer

We thank the reviewer for their constructive comments.

Reviewer's comments: (and our Responses in bold font)

Reviewer #1 (Remarks to the Author):

I carefully read the authors' reply to my comments and, although I appreciate they accepted my suggestion to include ToF-SIMS analysis, I also have to say that some statements in the section describing the results of molecular analysis still look inconclusive to me. Specifically:

Lines 93-97:

"The peak at 1033 cm⁻¹ presumably is a Si-O stretch mode from microcrystallites of a silicate phase within the fossil. Absorption due to the amide II band, at 1550 cm⁻¹ in the reference spectra, may be present within the broad elevated region of absorbance in the fossil tissue, and may even be slightly shifted to lower wavenumbers and thus contribute to the peak visible at ~1510 cm⁻¹."

Arguments phrased using expressions like: "presumably", "may be present" and "may even be slightly shifted" look more speculative than conclusive to me. They look to me more based on personal interpretation rather than strong evidence.

See final statement in this response; but yes, those qualifiers were intentionally included to tone down the interpretations that readers might misconstrue as unquestionably conclusive.

Also the interpretation of the ToF-SIMS results is tentative. In lines 131-135 the authors claim: "The peak at 275.16 amu in the negative spectrum is approximately consistent with a tetrapeptide –Ser-Gly-Gly-Gly- with a mass of 276.13 amu ($C_3H_7NO_3 + 3C_2H_5NO_2 - 3H_2O = C_9H_{16}N_4O_6$), while the peak at 413.26 amu in the negative spectrum accords roughly to a tetrapeptide –Thr-Thr-Pro-Pro- with a mass of 414.24 amu ($2C_4H_9NO_3 + 2C_5H_9NO_2 - 3H_2O = C_{18}H_{30}N_4O_7$)."

The identification of short peptides by "approximately" or "roughly" matching them to ToF-SIMS masses doesn't seem rigorous enough to me. If we then look at the suggested sequences: Ser-Gly-Gly-Gly and Thr-Thr-Pro-Pro we don't see the G-X-Y pattern typical of collagen: the most abundant proteins originally present in the fossilised tissue and arguably the one that has higher chances to be recovered. In the authors opinion, to which proteins do the identified peptide sequences belong to?

The common motifs in collagen are glycine-proline-X and glycine-X-hydroxyproline. Besides these amino acids, there are around 10 more common to collagen. Proline is the second most abundant amino acid in skin collagen after glycine, hydroxyproline is fourth, and serine, lysine, and leucine come in 8th, 9th, and 10th respectively. Threonine is 12th.

The reviewer raises an important point with respect to the ToF-SIMS. We have completely changed this paragraph based on a consultation with a ToF-SIMS protein analysis expert (Dr. Nick Lockyer, added to acknowledgements). Indeed, the mass at ~276 amu that maps with the FTIR amide bands is consistent with a Gly-X-Y moiety and this is much more probable than what we originally proposed in light of the fact that the organic residue is most likely derived from collagen breakdown. The text now directly answers the reviewer's question, "To which proteins do the identified peptide sequences belong to?" The answer here is, "We have not identified peptide sequences. We have resolved and mapped a mass fragment distribution consistent with G-X-Y type residue from collagen." Furthermore, for several reasons the resolved mass at ~414 amu is problematic, and so we do not base any conclusions on that at this time.

In conclusion in my opinion the authors should add to their text a clear cautionary note indicating that all their interpretation of the molecular results are tentative and compatible with, but not conclusively proving, the preservation of ancient biomolecules.

We have done so, although generally we prefer "probable" rather than "tentative"- this is a subjective judgement call but the text above explains our logic. "Proof" might even be construed to be impossible as all science is tentative. Here is our qualifying statement:

"The results from the chemical analyses allow us to conclude that organic residue is correlated with the structures we have resolved. Our data are not sufficient to definitively resolve the source of this residue, however the analytical results are consistent with these organics being derived from a collagen precursor, thus inductively strengthening our morphological interpretation."